# Rethinking Environmental Flows for the Yellow River Estuary by Trading Off Crop Yield and Ecological Benefits

**Aiping Pang** [1,2], **Fen Zhao** [1], **Chunhui Li** [1,*] and **Yujun Yi** [1]

[1] Key Laboratory for Water and Sediment Science of the Ministry of Education, School of Environment, Beijing Normal University, Beijing 100875, China; qinglan231@163.com (A.P.); zhaofen@mail.bnu.edu.cn (F.Z.); yiyujun@bnu.edu.cn (Y.Y.)
[2] Department of Public Management, Nanjing Academy of Administration, Nanjing 210046, China
* Correspondence: chunhuili@bnu.edu.cn

**Abstract:** To solve the water use conflicts between agriculture and ecosystems in arid and semiarid areas, a multi-objective trade-off analysis method was applied to determine the environmental flows (e-flows) for the Yellow River Estuary, by considering the temporal and spatial discrepancies in water allocation. The results showed that during average years, a loss of $3.7 \times 10^8$ yuan was caused with every $1 \times 10^8$ m$^3$ of e-flows under the baseline scenario. The crop growth stages of April–July are sensitive periods for water requirements, and over 5000 yuan/ha production losses were caused by prioritizing e-flows during this time in dry years. The stages from July–October require more water by ecosystems than other stages, and the recommended e-flows during this time accounted for 57% of the e-flows during the total year. Under scenarios 1–3, which represent the short-term, medium-term and long-term scenarios, more water resources were supplied by underground water and water diversion projects; however, alleviating the water use contradiction remained difficult in dry years. During average years, e-flows between 148 and $168 \times 10^8$ m$^3$ are recommended to meet the ecological objectives of survival, reproduction and biological integrity of species for the Yellow River Estuary. The recommended e-flows in wet years could meet higher ecological objectives but still barely achieve the targets of sediment transport and ecosystem dynamic balance. In dry years, the economic losses may be beyond the acceptance of irrigation stakeholders if more water is allocated to improve e-flows. In this case, $71 \times 10^8$ yuan would be paid to them to compensate for their losses. This study proposes an e-flow recommendation framework that is economically and ecologically optimal in areas with irreconcilable water-use contradictions.

**Keywords:** environmental flows; water-use efficiency; temporal and spatial discrepancies; trading off; the Yellow River Estuary

## 1. Introduction

An increasing number of water resources have been used for social and economic development, which has led to continuous declines in water availability for freshwater habitats, such as riparian floodplains, wetlands, and estuaries [1]. The diversion of water to irrigated agriculture is one important reason for the serious decline of freshwater ecosystems in many basins [2]. Globally, 70% of water resources are allocated to irrigation areas via water conservancy projects, such as reservoirs and dams. The interception process changes the natural properties of rivers and causes ecological problems [3]. E-flows are the basis for maintaining the health and sustainability of ecosystems during exploitation of water resources [4]. Nevertheless, most approaches to determine e-flows remain grounded in biophysical science [5].The traditional methods for e-flow evaluation are based mainly on the protection of certain ecological objectives, including the maintenance of the natural hydrological regime and the protection of typical biological resources or natural habitats [6]. Research on e-flows based on ecological objectives provides a quantitative standard for the maintenance of ecosystem health. However, under the ecological objective-based

method, e-flow evaluation results are also difficult to apply for water resource allocation schedules because upstream economic losses always exceed the acceptable range for stakeholders [7–9]. Well-functioning water allocation policies should not only be economically efficient but also socially acceptable, to reduce the likelihood of failure of water reallocation to the environment [10].

Fifty-two percent of irrigated areas around the world will face a loss of more than 10% of agricultural production after guaranteeing e-flows. The production loss caused by guaranteeing e-flows is quite shocking, considering that irrigation water can only support 15% of the total production [11]. Irrigation areas play an important role in guaranteeing social stability and food security [12]; however, the interlinkages between water for irrigation and for ecosystems are complex [13], and the water-use contradiction between agriculture and ecosystems is always irreconcilable in arid and semiarid areas. The long-term sustainability of water resources requires the ability to transparently assess trade-offs and feedbacks between outcomes related to ecosystem integrity and human well-being [14]. Studies have been conducted to evaluate economic and ecological outcomes of water allocation. For example, Cai (2004) used scenario analysis to evaluate the trade-offs of water use between agriculture and ecosystems, with the background of the South-to-North Transfer Project [15]. The Water Evaluation and Planning System Version 21 was previously integrated with a water resource management process to analyze water allocation–related issues [16]. Similarly, Pang et al. (2014) developed a conflict analysis framework that could be used to determine e-flows after balancing water requirements for ecosystem protection and irrigation processes [17]. Furthermore, some researchers have tried to optimize decisions about water allocation by using a probabilistic graphical model [18,19]. These studies provided feedback on the best water distribution schedule to protect ecological objectives, thus providing a rational basis for the recommendations of e-flows and allocation of water resources for different stakeholders. However, the influence of temporal and spatial discrepancies in the water allocation process has received little attention thus far, which may reduce the water distribution potential, especially in arid and semiarid areas [20]. Exploring these factors could help us to provide the best solutions for managing e-flows.

The objective in this study was to develop a multi-objective trade-off analysis model that could be used to recommend e-flows after balancing water-use efficiency between agriculture and ecosystems. It comprises three modules that (1) analyze the temporal and spatial discrepancies of water use by different stakeholders; (2) evaluate the water-use efficiency to assess the risks faced by crop production and ecological protection objectives under different water allocation schedules; (3) recommend reasonable e-flows under the scope of economic loss acceptability while maintaining a higher level of ecological objectives.

## 2. Methods

### 2.1. Study Area

The Yellow River is the largest river that flows through Shandong Province and is the main water source for the Shandong irrigation area. It enters Shandong Province from Dongming County, passes through 25 counties, and finally flows into the Bohai Sea (Figure 1). The Yellow River Estuary is located in eastern Shandong province. The freshwater wetland area of the estuary is 792.7 km$^2$ [21]. Since the 1970s, the contradictions between different water-use stakeholders, especially agriculture and ecosystems, have become a restrictive factor for economic development and ecosystem protection. Because the limited water resources of the Yellow River cannot meet the increasing needs of different stakeholders, the lower reaches of the Yellow River had frequent discontinuations in the late 20th century. The reduction of the runoff decreases the sediment transport to the sea, changes the sediment dispersal pattern at the estuary [22], and also causes ecological problems in the Yellow River Estuary.

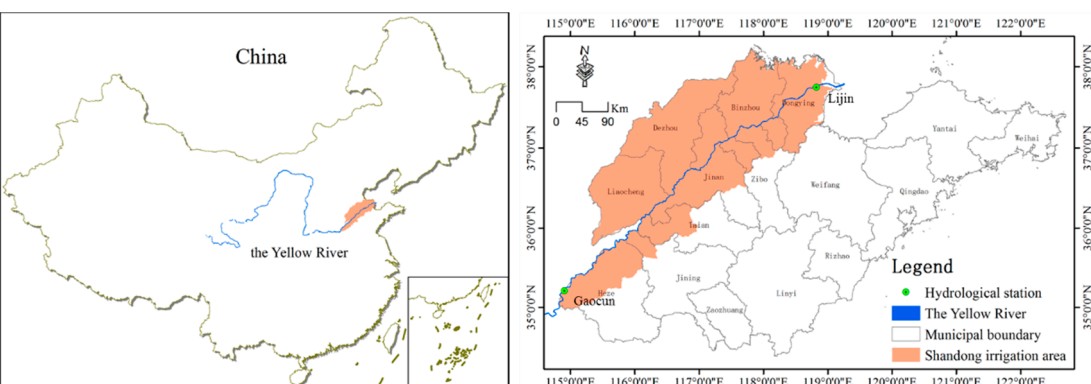

**Figure 1.** Location of the study area.

In 1999, an integrated water regulation schedule was applied by government. Rotation irrigation plans and water current limiting measures have generally been applied in irrigation processes to avoid drying up the Yellow River. Although the regulation schedule did not cause widespread crop failure, quite a few farmlands were not irrigated on time or suffered under erratic irrigation frequencies.

The Shandong irrigation area is located between Gaocun hydrological station and the Yellow River Estuary and is one of the most important economic development zones and crop production areas in China (Figure 1). The Shandong irrigation division projects were constructed starting in the 1950s; the irrigation area increased rapidly from $3.5 \times 10^4$ ha to $60.8 \times 10^4$ ha within 10 years and stabilized at $194.9 \times 10^4$ ha after 2000 (Figure 2A). The proportion of the Yellow River basin in Shandong Province only accounts for 1.36% of the total basin area, and among all the provinces (regions) where Yellow River water resources are supplied, Shandong Province accounts for the highest proportion at 22.4% [23]. The Gaocun hydrological station is the first station where the main stream of the Yellow River flows into Shandong Province, and its monitoring results represent the amount of Yellow River water resources used by different water-use stakeholders. The Lijin hydrological station is located in the estuary of the Yellow River, and its monitoring value represents the amount of freshwater resources entering the wetland ecosystem. From the 1950s to the 1970s, rainfall resources could basically meet the needs of water requirements for human welfare in Shandong Province, and runoff monitored at the Gaocun and Lijin stations could maintain almost the same level. Since the 1980s, with the development of industry and agriculture, the need of water resources by humans has become increasingly high. The differences between the runoff in Gaocun and Lijin Stations has shown an increasing trend year over year (Figure 2B).

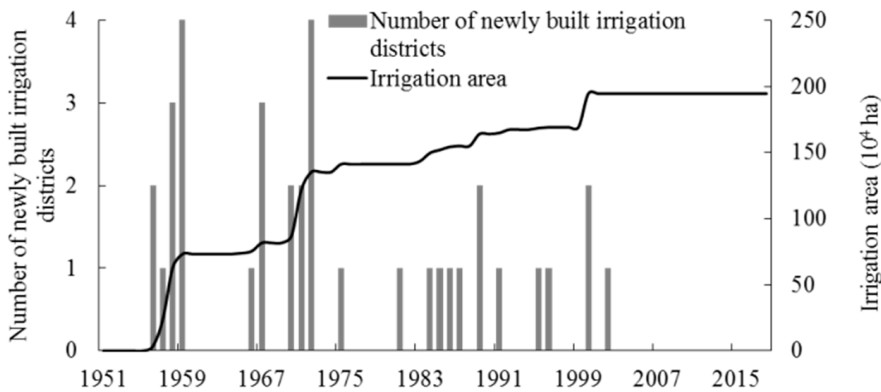

**(A)**

**Figure 2.** *Cont.*

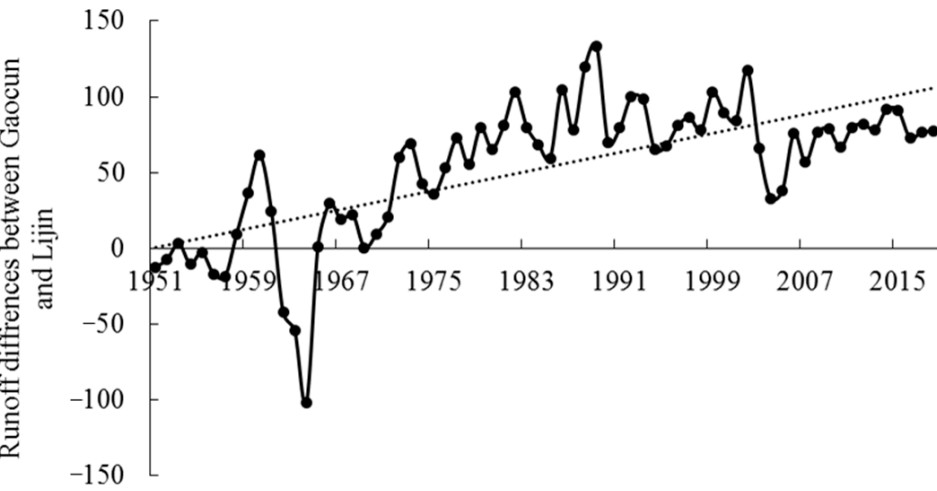

**(B)**

**Figure 2.** The number of newly built irrigation districts and the annual irrigation area (**A**) and the runoff differences between Gaocun and Lijin hydrological stations (**B**) from 1951–2018.

### 2.2. Methods

The multi-objective trade-off analysis method was used to recommend the e-flows for the Yellow River Estuary, which includes three modules (Figure 3).

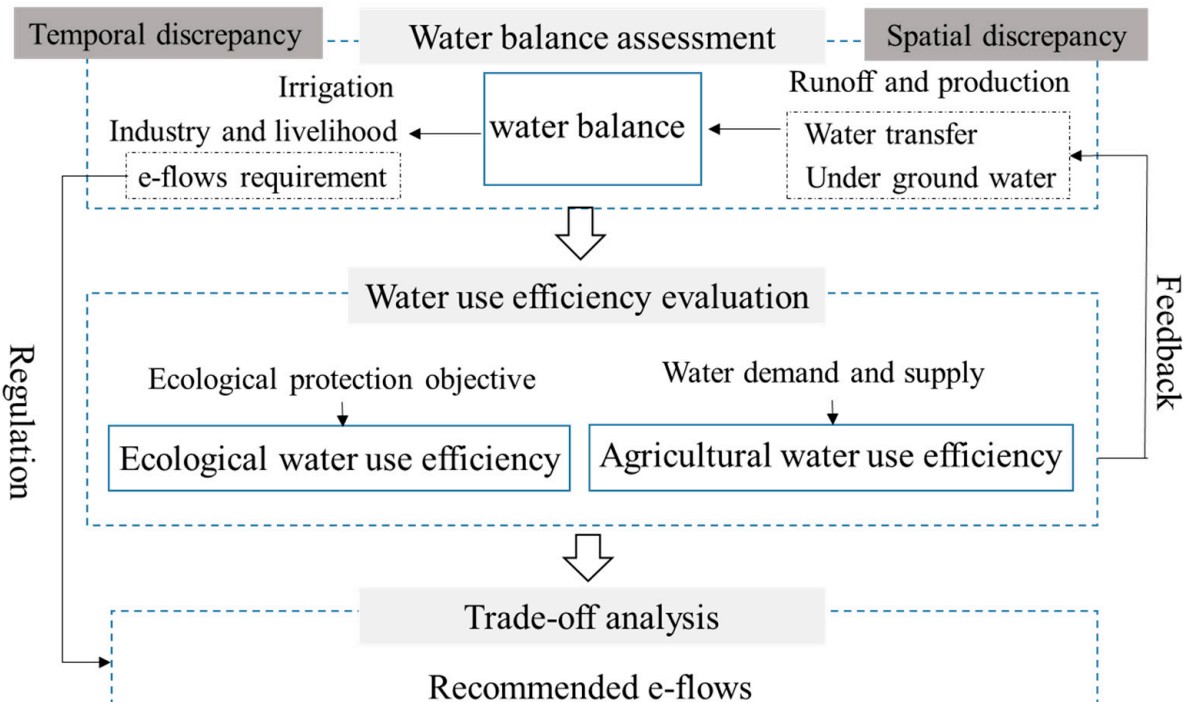

**Figure 3.** Module structure for the multi-objective trade-off analysis model.

Module 1 is the water balance assessment module. It distributes water resources for different stakeholders monthly, and further evaluates water supply and demand conditions for different crop growth stages. To reflect the accurate water requirements by agriculture, spatial discrepancies in rainfall and runoff for different irrigation areas are considered in the evaluation process.

Module 2 is the water-use efficiency evaluation module. The agricultural water-use efficiency is used to assess the risks faced by crop production under different water allocation schedules. Using the "accurate distribution" principle, the monthly allocation proportion for agriculture from the water transfer project and underground water are determined by feedback results from module 1. The ecological water use efficiency is used to assess the achievability of ecological protection objectives under different water allocation schedules.

Module 3 is the trade-off analysis module. A reasonable e-flow allocation scheme is evaluated under the conditions of economic acceptability of agricultural production losses while maintaining a higher level of initial e-flows.

### 2.2.1. Water Balance Assessment

Under average hydrological conditions, the agricultural water demand can be basically met by building reservoirs and water diversion projects. The interception and overutilization of water resources have caused the degradation of downstream ecosystems. The protection of ecosystems has become a millennium project in China, and the improvement of estuary health requires the control of human occupation of upstream water resources. In this study, irrigation water availability was determined based on the water balance principle, which takes the maintenance of e-flow requirements and industrial and domestic water use as priorities.

$$W_{ir}^a = \begin{cases} W_{in} + \alpha S_T P + W_u + W_t - W_d - W_i - W_e & W_{in} + \alpha S_T P + W_u - W_d - W_i - W_e < W_a^r \\ W_a^r & W_{in} + \alpha S_T P + W_u - W_d - W_i - W_e \geq W_a^r \end{cases} \quad (1)$$

where $W_{ir}^a$ is the irrigation water availability (m$^3$), $W_{in}$ is the runoff from the upstream (m$^3$), $\alpha$ is the water production coefficient (dimensionless), $S_T$ is the watershed area (m$^2$), $P$ is the annual precipitation (m), $W_u$ is the groundwater extraction (m$^3$), $W_t$ is the water resources from the transfer project (m$^3$), $W_d$ is the domestic water requirement (m$^3$), $W_i$ is the industrial water requirement (m$^3$), $W_e$ is the e-flow requirement, which is calculated based on the protection of different ecological objectives (m$^3$), and $W_a^r$ is the agriculture water requirement (m$^3$).

The e-flow requirement, $W_e$, namely, the initial e-flows, refers to the amount of water needed by the ecosystem to maintain a healthy and stable state. The initial e-flows vary according to the different objectives for ecological protection, restoration or construction [24]. Sun et al. (2008) calculated the e-flow requirement for the Yellow River Estuary by adopting the ecological target integration method [21]. Variations in different types of water requirements, such as water and biological cycle maintenance and biological habitat conservation, were considered in this method, and then the total e-flow requirement with different grades was calculated by adding consumable and non-consumable water requirements according to the "additive" and "maximum" principles.

$$W_e = \sum_{i=1}^{n} W_i + Max(W_{j1}, W_{j2}, \ldots, W_{jm}) \quad (2)$$

where $W_i$ is the consumable water requirement (m$^3$), $W_j$ is the non-consumable water requirement (m$^3$), and $n$ and $m$ represent the number of consumable and non-consumable ecological protection objectives, respectively.

The agriculture water requirement, $W_a^r$, is calculated by considering the spatial discrepancies for different irrigation areas. An irrigation area may cover several administrative districts, and the same administrative district may contain more than one irrigation area. However, the implementation of Yellow River water distribution schemes cannot be separated from the overall coordination of administrators. According to the research of Pang et al. (2021) [25], irrigation areas in Shandong Province have been divided into 18 regulation districts. The Thiessen interpolation method and geostatistical analysis methods

within ArcGIS 10.2 were used to calculate $W_a^r$ for different districts. For a specific regulation district, $W_a^r$ can be determined as follows:

$$(W_a^r)_A = [(ET_m)_x R(S_p)_x + (ET_m)_y R(S_p)_y \dots + (ET_m)_n R(S_p)_n](S_p)_A \tag{3}$$

where $(W_a^r)_A$ is the agriculture water requirement for regulation district $A$ (m$^3$), $(ET_m)_n$ is the actual evapotranspiration for weather station $n$ (m), $R(S_p)_n$ is the proportion of the Thiessen Polygon area generated by weather station $n$ in regulation district $A$ (dimensionless), $(S_p)_A$ is the irrigation area for regulation district $A$ (m$^2$), and $x, y \dots n$ represent the weather stations located in the regulation district $A$ (dimensionless).

The actual evapotranspiration, $ET_m$, is determined by the potential evapotranspiration ($ET_0$) and the crop coefficient ($k_c$) [26]. $ET_0$ is calculated by the Penman Monteith Equation [27], and the calculation process is described in the research by Liu et al. (1997) [28].

$$ET_0 = \frac{\frac{p_0 \triangle}{p\gamma} + 0.26(E_s - E_a)(1 + cU_2)}{\frac{p_0 \triangle}{p\gamma} + 1} \tag{4}$$

where $p_0$ is the atmospheric pressure at sea level (kPa), $p$ is the actual atmospheric pressure (kPa), $E_s$ is the saturation vapor pressure (kPa), $E_a$ is the actual vapor pressure (kPa), $c$ is the correction coefficient of wind speed (dimensionless), $U_2$ is the wind speed at 2 m height (m/s), $\triangle$ is the slope of the saturation vapor pressure and temperature curve (kPa/°C), and $\gamma$ is the thermometer constant (kPa/°C).

2.2.2. Water-Use Efficiency Evaluation

1. Ecological water-use efficiency evaluation

The initial e-flows for maintaining ecosystem health are calculated to protect different ecological objectives, and values lower or higher than the e-flow threshold will inevitably lead to the degradation of the ecosystem [21]. Therefore, initial e-flows do not provide a definite value; however, they do provide boundary conditions for ecological objective protections [17]. The ecological water-use efficiency was represented by different initial e-flows levels and ecological objectives. Twenty different levels of initial e-flows were determined by the equipartition of high and low levels of the boundary for further scenario analysis. Within this boundary, the more water allocated to ecological needs, the more beneficial the ecological water use, and the healthier the ecosystem.

2. Agricultural water-use efficiency evaluation

The D-K model was proposed to evaluate the relationship between crop yield loss and evapotranspiration deficiency at different crop growth stages [29], and an improved D-K model [30] was applied to calculate the economic losses associated with agricultural water shortages. We adopted the following equations to evaluate the agricultural water-use efficiency in the Shandong irrigation area, associated with securing e-flows for the Yellow River Estuary:

$$Q = \sum_{j=1} \sum_{y=1} q_{km}^j S_j [1 - k_y (1 - \frac{(W_{ir}^a)_y}{(W_a^r)_y})]_j \tag{5}$$

$$L = \sum_{j=1} \sum_{y=1} q_{km}^j S_j [k_y (1 - \frac{(W_{ir}^a)_y}{(W_a^r)_y})]_j \tag{6}$$

where $Q$ is the crop output value (yuan), $L$ is the production loss (yuan), $q_{km}^j$ is the maximum yield production without water stress (t/ha), $S_j$ is the planting area (ha), and $k_y$ is the yield response factor (dimensionless), all for crop $j$ in crop growth stage $y$.

### 2.2.3. Trade-Off Analysis

A reasonable e-flow allocation should meet the basic needs of ecological health, which is within the range of high and low limits of initial e-flows, while also meeting the needs of upstream economic development as much as possible. However, with the improvement of e-flow levels, the total crop output value continues to fall. The following two conditions are set in the trading-off process (Figure 4).

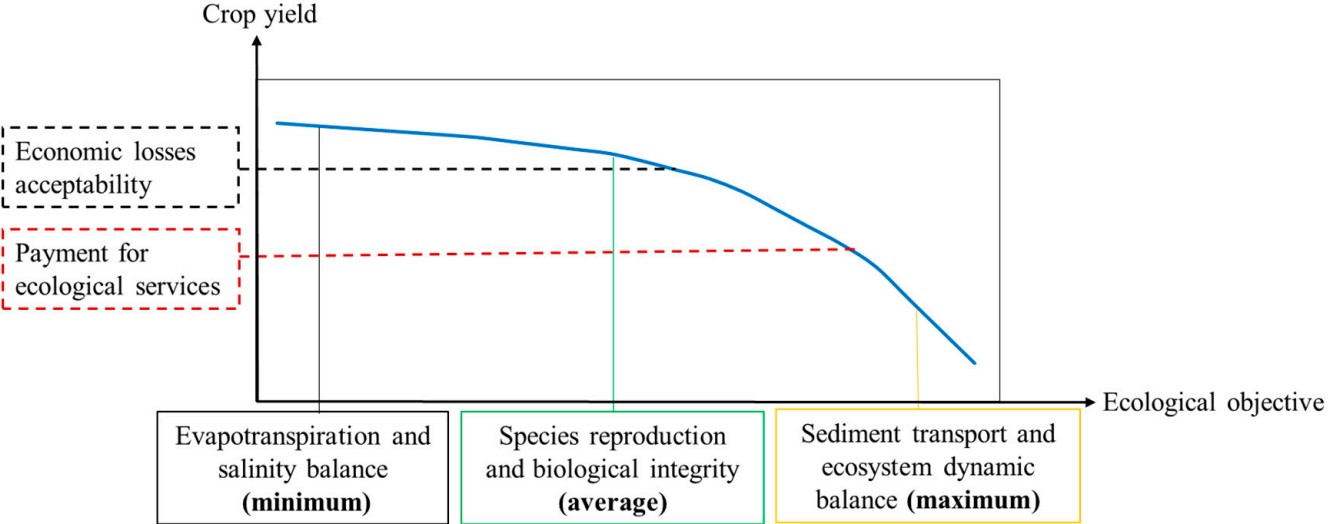

**Figure 4.** Trade-off analysis between crop yield and ecological benefits.

Economic losses acceptability: Farmers are willing to give up part of their water rights to maintain the e-flows, within the limits of acceptable economic losses. The government of China implemented an agricultural tax policy before 2006, and the formulation was based on the farmers' acceptance of economic losses. Under this policy, farmers have to pay 15% of their agricultural income as an agricultural tax. This standard is used as economic losses acceptability for farmers for giving part of their water rights.

Payment for the ecological services (ecological compensation) standard: If production losses exceed the range of farmers' acceptability, then there is an irreconcilable contradiction between agricultural and ecological water use. Thus, we could either guarantee agricultural production at the expense of ecological health or adopt the "buy-back" form to compensate the losses incurred by maintaining e-flows.

### 2.3. Data Source and Scenario Setting

#### 2.3.1. Data Source

Crop planting data for the Shandong irrigation area was obtained from the study of Pang et al. (2021) [25]. Daily meteorological data from 1951 to 2018 (including longitude and latitude, altitude, atmospheric pressure, temperature, humidity, precipitation, wind speed, sunshine time, etc.) were used to calculate the potential evaporation for the 15 weather stations (Chaoyang, Dezhou, Dingtao, Dongying, Heze, Huimin, Jinan, Kenli, Linxian, Taian, Taishan, Yanzhou, Zhangqiu, Zibo and Zichuan); this information came from the China Meteorological Data Service Center (http://data.cma.cn).

The potential evapotranspiration and precipitation were calculated on a daily scale and then accumulated to a monthly scale. Crop coefficient data for crops planted in the study area were obtained from the research of Chen et al. (1995) [31]. The Thiessen interpolation method and geostatistical analysis method within ArcGIS10.2 were used to calculate agricultural water needs for different districts.

The runoff data at the Gaocun and Lijin hydrological stations were obtained from the research of Pang et al. (2014) [17] and supplemented by the Yellow River Water Resources Bulletin (YRCC, 1998–2018) [23]. According to the preliminary evaluation of

water resources in Jinan City [32], the water production coefficient was 0.378. In recent years, the average amount of exploitable underground water resources in the Shandong irrigation area was approximately $40 \times 10^8$ m$^3$ per year [33,34].

The industrial and domestic water requirements data from 1998–2017 were obtained from the Yellow River Water Resources Bulletin [23]. There is no recorded official data for the years 1951–1997 and 2018. We supplemented the missing data by multiplying the proportion coefficients by the available water resources. The average proportion coefficient was 7.4% for the years 1951–1997 and 15.7% for the year 2018 [23]. The available water resources were the differences of runoff data monitored at Gaocun and Lijin hydrological stations.

### 2.3.2. E-Flow Boundary Setting

Selecting the appropriate ecological protection objectives and determining the reasonable initial e-flow range will directly affect the recommendation of e-flows and the corresponding economic losses of the agricultural sector. In this study, the initial e-flow boundary adopted the assessment results of Sun et al. (2008) [21]. The minimum e-flow was $134.21 \times 10^8$ m$^3$, which could meet the targets of estuary evaporation consumption and salinity balance. The average e-flow was $162.73 \times 10^8$ m$^3$, under which the survival and reproduction of the target species could be guaranteed. If the actual e-flows are less than this value, then biological reproduction conditions will be destroyed and biomass and biological integrity will be reduced. The maximum e-flow was $274.91 \times 10^8$ m$^3$; maintaining this value is beneficial to sediment transport and the balance of the whole ecosystem.

### 2.3.3. Scenario Setting

South-to-North Water Diversion project: The first phase of the eastern route of the South-to-North Water Diversion Project was completed in 2013. The main objective of the project was to deliver water resources for industrial or urban domestic use, to provide agricultural and ecological water use when needed, and to reduce the dependence of the irrigation area on the Yellow River and groundwater. Since the completion of the project, a total of $39.21 \times 10^8$ m$^3$ Yangtze River water has been diverted to Shandong Province. According to the 2019–2020 Water Transfer Plan approved by the MWR (Ministry of Water Resources), the South-to-North Water Diversion Project transferred $7.03 \times 10^8$ m$^3$ of water to Shandong Province in the first phase. Approximately $7.03–12.23 \times 10^8$ m$^3$ of water will be transferred to Shandong Province as planned in the second phase. Thus, the water transfer parameters were set as 7.03, 9.63 and $12.23 \times 10^8$ m$^3$, which represent the short-term, medium-term and long-term scenarios, respectively.

Groundwater exploitation: The average amount of exploitable underground water resources in the Shandong irrigation area is approximately $40 \times 10^8$ m$^3$ per year [33,34]. These data are regarded as a reasonable level that can maintain the health of groundwater ecosystems and can also ensure the safety of agricultural water availability. River drying is common in the area north of the Yellow River, and the available water resources are decreasing every year. In reality, groundwater is frequently over-extracted to alleviate water use conflicts, which results in geological disasters, such as ground subsidence, collapse and cracks. Thus, the extraction amount of groundwater was set as 40, 60 and $80 \times 10^8$ m$^3$ per year, which represent the situations with 0, 50 and 100% over-extraction of groundwater.

Based on the above water allocation and exploitation schemes, we established 10 scenarios (Table 1). In the baseline scenario, the total available water resources for the Shandong irrigation area only came from upstream inflow and surface runoff produced by precipitation. The calculation results from the baseline scenario provide the monthly water allocation ratios for the other nine scenarios.

**Table 1.** Parameter settings for different scenarios.

| Scenarios | Water Resources Allocation Scheme ($10^8$ m$^3$) | |
|---|---|---|
| | **Groundwater Exploitation** | **South-to-North Water Diversion Project** |
| Baseline | 0 | 0 |
| 1 | | 7.03 (short-term) |
| 2 | 40 (reasonable) | 9.63 (medium-term) |
| 3 | | 12.23 (long-term) |
| 4 | | 7.03 (short-term) |
| 5 | 60 (50% over-exploited) | 9.63 (medium-term) |
| 6 | | 12.23 (long-term) |
| 7 | | 7.03 (short-term) |
| 8 | 80 (100% over-exploited) | 9.63 (medium-term) |
| 9 | | 12.23 (long-term) |

## 3. Results and Analysis

### 3.1. Water-Use Efficiency under the Baseline Scenario

Crop production in the irrigation area varied under different e-flow allocation plans. In dry years, if the initial e-flows could meet the ecological objective of estuary evaporation consumption and salinity balance, we should allocate $134.21 \times 10^8$ m$^3$ water to the Yellow River Estuary. The total crop output value was $227 \times 10^8$ yuan, which accounted for 36.74% of the potential crop output value (maximum yield production without water stress) for the Shandong irrigation area. This ratio was reduced to 23.52% when the e-flows could meet the ecological objective of target species survival and reproduction. There was no water left for the irrigation process when the e-flows allocated were higher than $190 \times 10^8$ m$^3$. In average years, with the improvement of the initial e-flow security level, the total crop output value also showed a trend of straight decline, a loss of $3.7 \times 10^8$ yuan was caused by ensuring every $1 \times 10^8$ m$^3$ of e-flows. In wet years, the contradiction between ecological and agricultural water use was alleviated. When the initial e-flows gradually increased from the lowest level to $180 \times 10^8$ m$^3$, the total crop output value could be maintained above $400 \times 10^8$ yuan. Then, with the improvement of the e-flow security level, the total crop output value dropped sharply (Figure 5A).

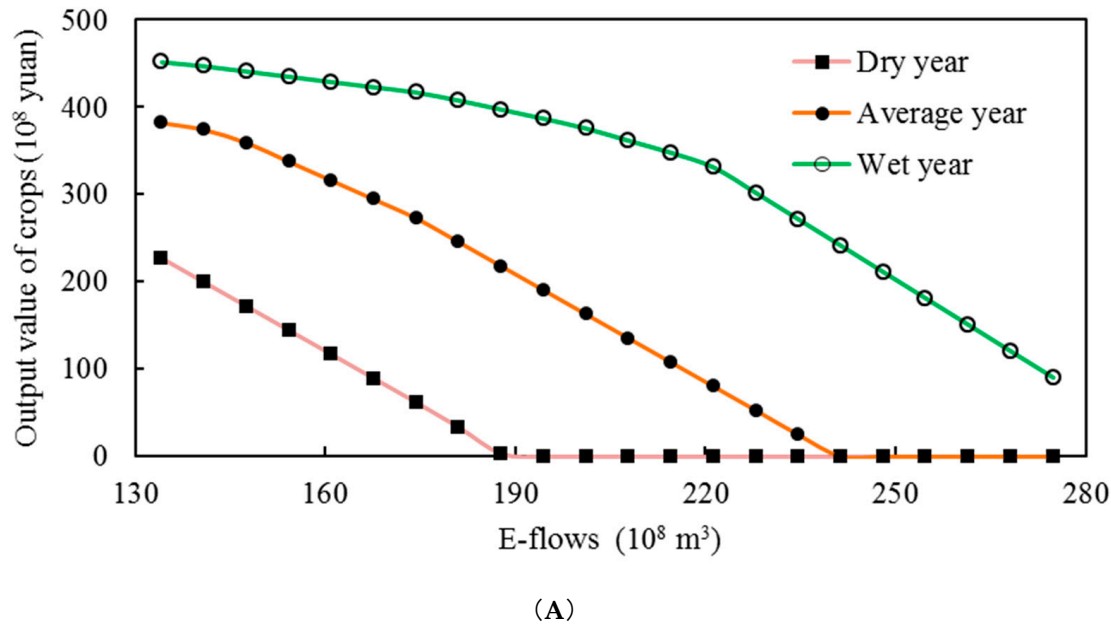

(**A**)

**Figure 5.** *Cont.*

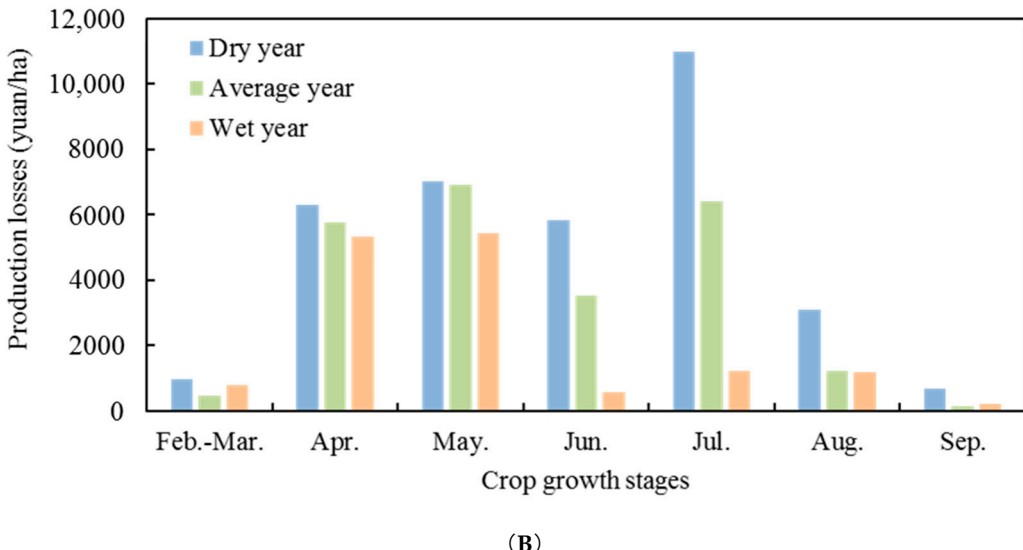

(**B**)

**Figure 5.** The relationship between e-flows for the Yellow River Estuary and the crop output value for the Shandong irrigation area (**A**) and the impact of e-flow security on production losses at different crop growth stages (**B**).

The sensitivity of crops to water at different growth stages was different; therefore, the guaranteed e-flows in different months had significantly different impacts on the agricultural production losses (Figure 5B). During the growing period of wheat from February to May and the growing period of corn in September, the guaranteed e-flows had little impact on agricultural production, and the average loss was less than 1000 yuan/ha. From April to July, crops are sensitive to water demand, especially in dry years, and the production loss based on e-flow security was greater than 5000 yuan/ha.

### 3.2. Water-Use Efficiency under Different Water Resource Allocation Scenarios

From the analysis of water-use efficiency under the baseline scenario, it can be seen that the securing of e-flows for the Yellow River Estuary has a huge impact on production in the Shandong irrigation area. It is difficult to relieve water-use conflicts between agriculture and ecosystems that only rely on upstream Yellow River inflow and surface water production. We analyzed the combined effect of groundwater exploitation and water diversion projects to solve the problem of water shortage. The monthly water allocation ratio was obtained based on the feedback of the baseline scenario (Figure 7A). Under scenarios 1–3, the groundwater utilization and water transfer from the South-to-North Water Diversion Project were allocated within a reasonable range. According to the trade-off analysis principle of e-flow determination, if the total crop output value is less than $522 \times 10^8$ yuan, the economic losses could not be accepted by irrigators. In dry years, in scenarios 1–3, the total crop output value under all water allocation schemes was less than $451 \times 10^8$ yuan; thus, this cannot provide e-flows within the acceptance of irrigators. In average years, the range of recommended e-flows was $148–168 \times 10^8$ m³. Within this range, ecological objectives, such as evaporation consumption, salinity balance, species survival and reproduction, and biological integrity, could be achieved in the Yellow River Estuary. In wet years, the recommended e-flow range was increased to $228–235 \times 10^8$ m³, which could meet higher ecological objectives. However, certain objectives were more difficult to achieve, such as sediment transportation and the dynamic balance of the whole ecosystem (Figure 7B).

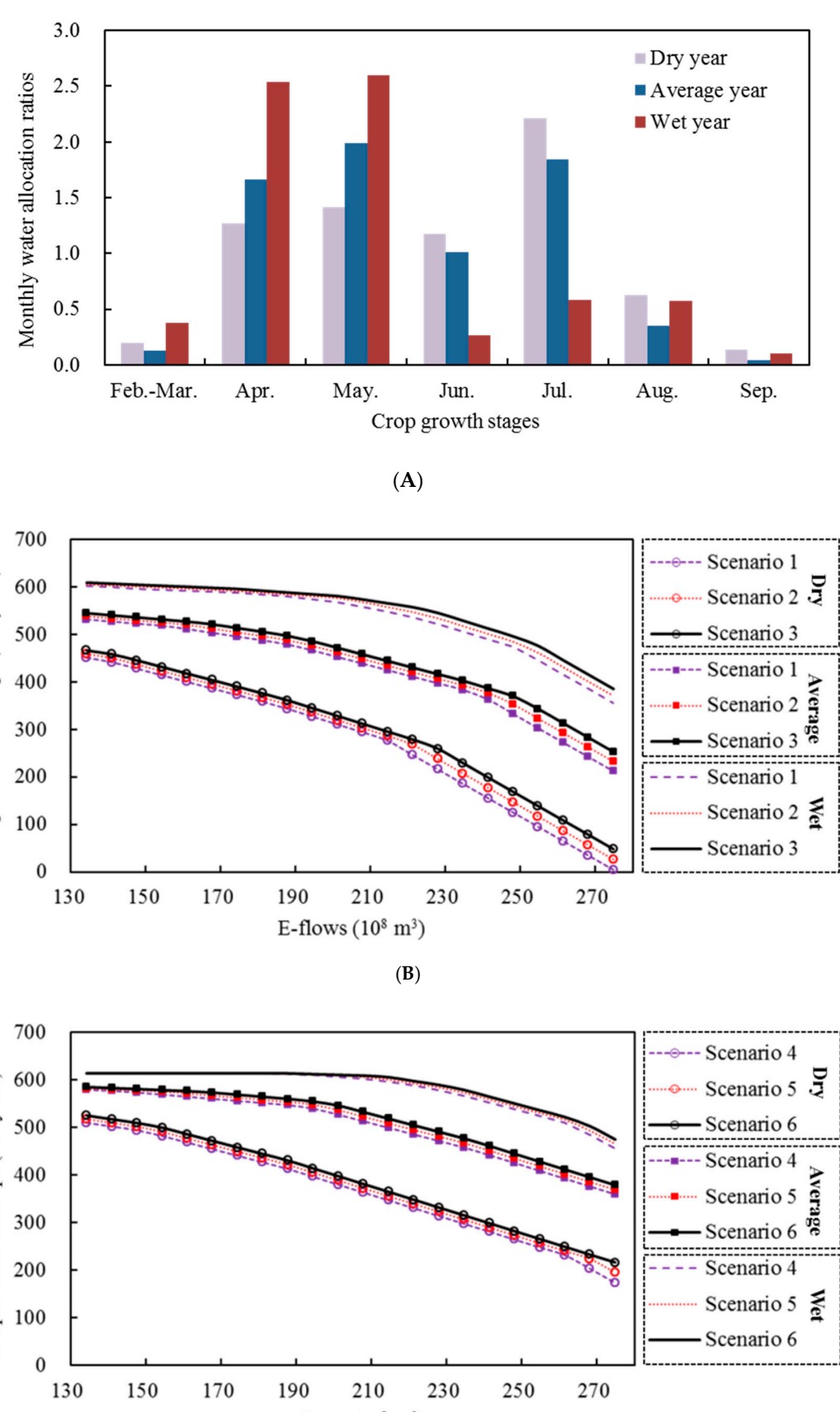

**Figure 6.** Cont.

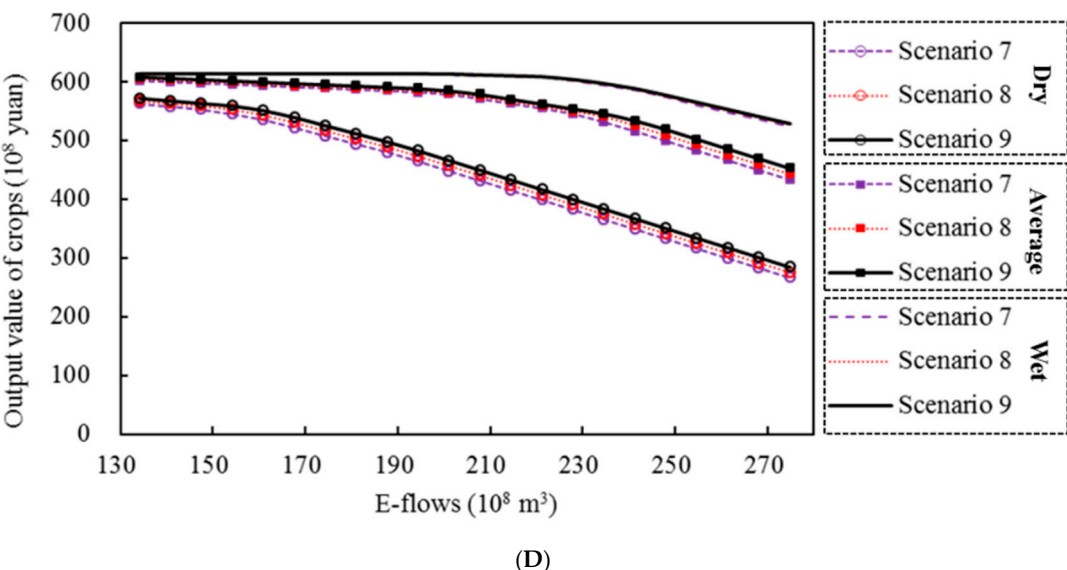

(**D**)

**Figure 7.** Water allocation ratios for different months (**A**) and the relationship between e-flows for the Yellow River Estuary and the crop output value for the Shandong irrigation area (**B**–**D**).

Under scenarios 4–9, the groundwater was over-exploited by 50% and 100%. This is a common situation in reality. Generally, farmers in the irrigation area use well irrigation to compensate for the scarcity of the Yellow River water. Under scenarios 4–6, the lowest level of e-flows could be guaranteed; that is, the ecological objectives of evaporation consumption and salinity balance in the Yellow River Estuary could barely be maintained. In average years, the e-flow range was $201–208 \times 10^8$ m$^3$; after exceeding this standard, the total crop output value declined sharply. In wet years, the e-flow range was $254–261 \times 10^8$ m$^3$, within the acceptance of economic losses (Figure 7C). Under scenarios 7–9, the maintenance of e-flows could meet the water demands for the survival and reproduction of the target species in different hydrological years. In wet years, the e-flows could meet the water needs for the higher ecological objective of sediment transportation and the whole ecosystem dynamic balance (Figure 7D).

*3.3. Recommended E-Flows of the Yellow River Estuary*

Scenarios 4–9 all involve the problem of groundwater over-exploitation. This is a common phenomenon in reality and can relieve water-use conflicts to a certain extent. However, the behavior of sacrificing groundwater health causes damage to the groundwater ecological environment. Therefore, in the water allocation plans, only the rational use of groundwater is considered, which is the e-flow allocation scheme under scenarios 1–3 (Figure 8). Under different e-flow allocation schemes (Figure 8A), the economic losses in dry years were all above 24%, which was beyond acceptability. It was impossible to maintain the ecological health of the estuary while ensuring the safety of agricultural production. Within the range of economic losses acceptability, the recommended e-flows only accounted for approximately 60% of the minimum level of initial e-flows. In average years, the recommended e-flows ranged from $148–168 \times 10^8$ m$^3$ under different scenarios. If the long-term water transfer volume of the South-to-North Water Transfer Project could be guaranteed, the e-flows for the survival and reproduction of the target species in the Yellow River were barely maintained. The e-flows under different scenarios in the wet years ranged from $228–235 \times 10^8$ m$^3$, which could meet a higher ecological objective. Judging from the monthly e-flows (Figure 8B), the water-use conflicts were most fierce during the corn growing period. That is, July to October was the peak period of ecological water needs, during which the recommended e-flows accounted for 57% of the total value of the whole year. December to April of the following year was the low period for ecological water needs, during which e-flows only accounted for 20% of the total volume of the year.

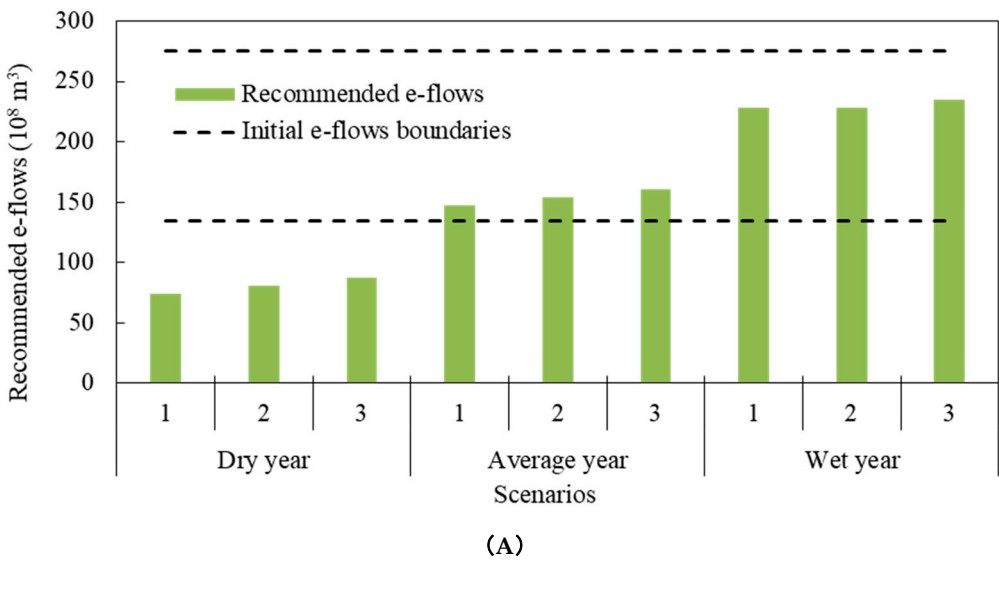

（A）

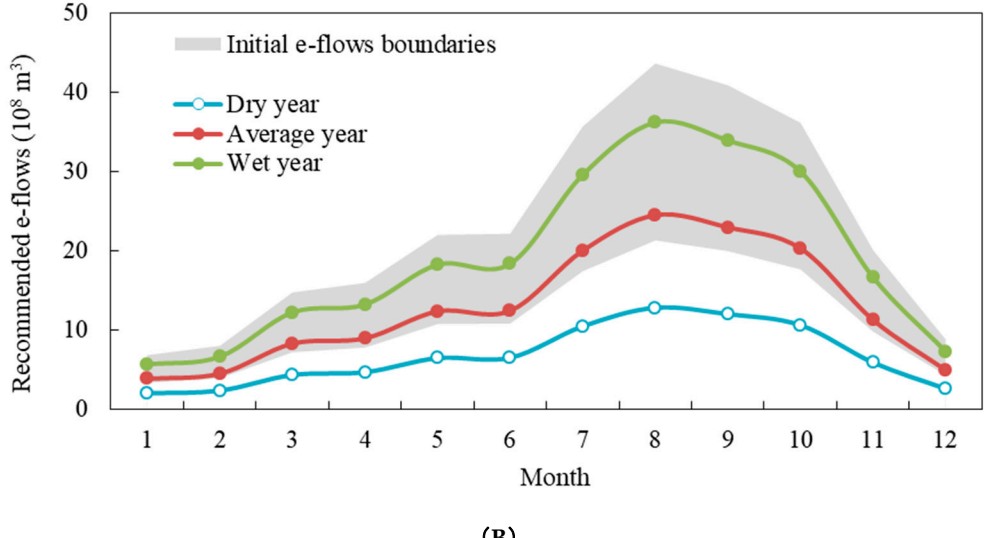

（B）

**Figure 8.** The recommended e-flows in scenarios 1–3 (**A**) and the monthly recommended e-flows for different hydrological years in scenario 1 (**B**).

## 4. Discussion

### 4.1. Evaluation of Food Security and Implementation of Ecological Compensation

The arable land and population within the Shandong irrigation area account for 58% and 48% of the province's total arable land and population, respectively. The Yellow River water resources are the lifeblood of agriculture along the Yellow River. The e-flow maintenance has a great impact on the food security of Shandong Province. In 2018, the total crop output value of the Shandong irrigation area was approximately $600 \times 10^8$ yuan, accounting for 52% of the province's total [35]. In scenario 1, the proportion of crop output loss was between 0.27 and 0.99, after securing different levels of e-flow in dry years. That is, the guarantee of e-flows resulted in a total loss of $159–595 \times 10^8$ yuan. The proportion was between 0.13 and 0.65 in average years, reduced from 0.02 to 0.42 in wet years (Figure 7). In most cases, the recommended e-flows could be guaranteed within the boundary of initial e-flows, and agricultural economic losses could also be accepted by irrigation stakeholders. However, to achieve higher ecological protection objectives, some necessary economic measures must be adopted to offset the economic losses beyond the acceptable range.

Ecological protection objectives and hydrological conditions are the two factors influencing the economic compensation standard for farmers. To achieve the basic ecological objectives of estuary evaporation consumption and salinity balance, only $71 \times 10^8$ yuan needs to be paid in dry years. If the ecological objectives for the survival and reproduction of the target species are reached, then the ecological compensation standards for dry years and average years are $121 \times 10^8$ yuan and $10 \times 10^8$ yuan, respectively. And to achieve the higher ecological objectives of sediment transport and the balance of the whole ecosystem, the compensation standards for dry years, average years, and wet years are 517, 309, and $165 \times 10^8$ yuan, respectively (Figure 9).

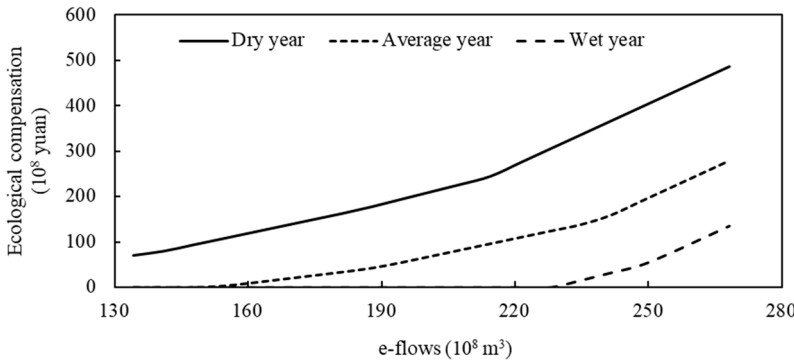

**Figure 9.** The relationship between e-flow security and ecological compensation.

### *4.2. The Influence of Temporal and Spatial Discrepancies in Water Allocation*

The protection of the ecosystem of the Yellow River has become an issue of great concern for the government of China. Many studies have focused on the evaluation of the water-use conflicts between the Shandong irrigated area and Yellow River Estuary [17,18]. In an area with a severe shortage of water resources, the temporal and spatial discrepancies for natural factors are critical issues that should be addressed in assessments. Specially, we combined these factors into our model structure in the trading-off process. The differences in the model structure, such as parameter settings, data scope and other factors, made it very difficult to compare methods used by different studies. The comparable parameters, such as the agriculture water requirement and irrigation water shortages caused by the ensuring of e-flows, were used in our modules and other trade-off analysis studies. They are the key factors for establishing water allocation schedules and are also very comparable. In 2010, the government of Shandong province established a local standard of irrigation quotas for the main crops in the Shandong irrigation area. Most of the studies used the data specified in the document as agriculture water requirements, thus neglecting the temporal and spatial discrepancies in water allocation. The irrigation water shortages could be reduced to $30–40 \times 10^8$ m$^3$, after ensuring different levels of e-flow by comparing this study and the research conducted by Pang et al. (2013) [30]. Using this "accurate water distribution" principle [25] does not produce a negative linear relationship between crop yield and ecological benefits (Figure 7). More water can be allocated to ecosystems without resulting in additional agricultural production losses caused by water shortages.

### 5. Conclusions

We developed a multi-objective trade-off analysis method to recommend the optimal e-flows for the Yellow River Estuary. We evaluated water-use efficiency between agriculture and ecosystems under different water allocation schemes. We found that higher amounts of water could be allocated for ecosystems without resulting in additional economic losses in the agricultural sector caused by water shortages. That is, we did not find a negative linear relationship between crop yield and ecological benefits when considering temporal and spatial discrepancies in water allocation. We determined the best point to maintain ecological objectives within the acceptance of economic losses.

Hydrological conditions are an important factor in determining recommended e-flows. In dry years, it is difficult to alleviate the contradiction between ecological and agricultural water use under the premise of rational water resource use. In average years, water is needed for evapotranspiration, salinity balance, species survival and reproduction, and biological integrity can be achieved through the short-, medium- and long-term water transfer processes of the South-to-North Water Transfer project. The guarantee of recommended e-flows in wet years can meet higher ecological objectives than in average years. However, achieving higher ecological objectives such as sediment transportation and ecosystem dynamic balance is more difficult.

This method allowed for an improved understanding of how seasonal and spatial variation in water allocation can impact agricultural processes and drive e-flow allocation. In addition, it can be used as an objective tool for determining acceptable e-flows that meet the needs of all stakeholders. Ecological compensation standards were also evaluated to achieve a higher ecological protection objective. Also, the method is broadly generalizable and allows for the incorporation of additional factors into analyses. For example, a user could investigate the impacts of processes like global climate change and increased human activity on water availability and demand.

**Author Contributions:** A.P. and F.Z. conducted the experiments, analysis and paper write-up, C.L. and Y.Y. offered guidance on the experiments, advised and modified the write-up. All authors reviewed the manuscript. All authors have read and agree to the published version of the manuscript.

**Funding:** This study is supported by the National Key Research and Development Program (2018YFC0407403 and 2017YFC0404401).

**Institutional Review Board Statement:** Not applicable.

**Informed Consent Statement:** Not applicable.

**Data Availability Statement:** The data presented in this study are available on request from the corresponding author.

**Acknowledgments:** We would like to extend special thanks to the editor and reviewers for insightful advice and comments on the manuscript.

**Conflicts of Interest:** The authors declare no conflict of interest.

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
