# Peer review of "Rethinking Environmental Flows for the Yellow River Estuary by Trading Off Crop Yield and Ecological Benefits"

_agriculture, doi:10.3390/agriculture11020116_

Round 1

Reviewer 1 Report

I couldn’t enjoy reading this bulky manuscript. There has been a lot done but presented in a messy way. Paper must be re-organized. E-flows is an important part of hydrology but I couldn’t find any hydrologic model used in the study.

Other concerns/comments:

-Abstract is really long..

-Research gap was not given in intro section. What has not been done? Please specify in the last two paragraph of the introduction. Research approach should be given after research questions/objectives.

-Discussion section doesn’t discuss/criticize/reflects about the paper results/methods. It just adds up extra subsections to the results. We should read “however” word several times in the discussion section. Also benchmarking with other manuscripts/results is necessary.

-Conclusion is only giving extra results. It should not contain much numbers/results but overall synthesis of the results.

Reviewer 2 Report

This paper uses a multiobjective balance method to solve the water use conflicts between agriculture and ecosystems and to determine the environmental flows (e-flows) of water for the Yellow River Estuary. These methods are applied for dry, average, and wet years. It also calculates the compensations paid for dry, average, and wet years when there is not enough supply of water. This is a noble work and the authors deserve congratulations for completing such a mammoth project.  However, it is hard to recommend this paper to publish in the present form because of the lack of clarity in many areas.  I am not able to justify the amount of Yuan to pay for different years and seasons. I also do not qualify to comment on the quantity of waters that are available in different years.  I am not a mathematician to argue with the algorithm used in this manuscript.

I looked at the clarity of the writing how a reader can perceive the real message after reading this paper. First thing, the paper needs to be reorganized and language needs refining and condensing. Second, please avoid the passive voice and long sentences. Where possible, please use the active voice and write short sentences. There are too many “respectively” in the write-up, but it is hard to know which belongs to which. For example, the sentence in lines 323 to 326.

Similarly, there are many run-on sentences, for example lines 19 to 22.

In some places, the amount of water needed is explained but not sure what is the aerial unit, for example lines 29-32.

What is the targets of sediment transport in line 32-34 referred to?    

In some places, compensation needed for flow maintenance is presented, but the explanation is very unclear, for example, lines 38-41.

Some typo errors, such as in line 103.

Different modules are presented, for example, lines 150, but such important parts are embedded in the middle of a paragraph, and readers may encounter a hard time finding them. How about making subtitles for different modules?

Some writings need further clarity, for example, line 253.

In line 255 -257: How about just simplifying

Thiessen interpolation and geostatistical analysis methods within ArcGIS 10.2 are used to calculate agricultural water needs for different districts.

Some sentences such as lines 304-306 are not clear.

What does “per year for years” refer to such as in line 314-315.

Many sentences are run on, for example lines 319-323, very confusing.

There are many run-on sentences that do not convey the real meaning.

Author Response

Reviewer 2:

Dear Editor,

Thanks a lot for your sincere comments and suggestions about the improvement of the manuscript. We all authors have talked about these comments carefully and accordingly revised the manuscript again. We have had the manuscript polished by a professional assistance in writing and carefully revised the issues the reviewer have mentioned. The point to point revisions for the comments listed as follows:

 Comment 1: This paper uses a multiobjective balance method to solve the water use conflicts between agriculture and ecosystems and to determine the environmental flows (e-flows) of water for the Yellow River Estuary. These methods are applied for dry, average, and wet years. It also calculates the compensations paid for dry, average, and wet years when there is not enough supply of water. This is a noble work and the authors deserve congratulations for completing such a mammoth project.  However, it is hard to recommend this paper to publish in the present form because of the lack of clarity in many areas.  I am not able to justify the amount of Yuan to pay for different years and seasons. I also do not qualify to comment on the quantity of waters that are available in different years.  I am not a mathematician to argue with the algorithm used in this manuscript.

Responds: Thanks for your comments and suggestions to improve my writings. We revised the manuscript, not limited to the comments, to make sure all the sentences could convey the real meanings.

 Comment 2: I looked at the clarity of the writing how a reader can perceive the real message after reading this paper. First thing, the paper needs to be reorganized and language needs refining and condensing. Second, please avoid the passive voice and long sentences. Where possible, please use the active voice and write short sentences. There are too many “respectively” in the write-up, but it is hard to know which belongs to which. For example, the sentence in lines 323 to 326.

Responds: the comments and suggestions are very helpful in improvement of the writings of our manuscript. We have had the manuscript polished by a professional assistance in writing and carefully revised the issues the reviewer have mentioned. We polished our manuscript sentence by sentence followed the questions suggested by the reviewer. For example, the sentence in Lines 320 to 326 in the original manuscript “and the total industrial and domestic water consumption accounted for 7.4%, 14.3% and 15.7% of the annual water diversion resources from the Yellow River by Shandong Province in the 1990s, 2000s and 2010s, respectively. The industrial and domestic water requirements from 1951-1997 and 2018 were determined by the annual average difference in runoff, which was monitored at the Gaocun and Lijin hydrological stations during the same period and multiplied by 7.4% and 15.7%, respectively.has been revised to “There is no recorded official data for the years 1951-1997 and 2018.  We supplemented the missing data by multiplying the proportion coefficients by the available water resources. The average proportion coefficient was 7.4% for the years 1951-1997, and 15.7% for the year 2018 [22]. The available water resources were the differences of runoff data monitored at Gaocun and Lijin hydrological stations.” in the revised manuscript in Lines 286 to 291.   

 Comment 3: Similarly, there are many run-on sentences, for example lines 19 to 22.

Responds: Thanks for your comments and suggestions. The sentence in Lines 19 to 22 in the original manuscript “The results showed that during average years, a loss of 370 million yuan was caused by ensuring 100 million m3 of e-flows when runoff from the Yellow River and rainfall were the only water resources for irrigation processes and ecosystems.has been revised to “The results showed that during average years, a loss of 3.7×108 yuan was caused by ensuring every 1×108 m3 of e-flows under the baseline scenario.” in the revised manuscript in Lines 15 to 17.    

 Comment 4: In some places, the amount of water needed is explained but not sure what is the aerial unit, for example lines 29-32.

Responds: Thanks for your comments. We are not quite understand the meaning of “aerial unit”. Do you refer the unit “billion m3in the original manuscript in Lines 29-32? I adopted the scientific notation method to represent the large numbers in the revised manuscript. For example “14.8 and 16.8 billion m3in the original manuscript in Lines30 has been revised to “148 and 168×108 m3in the revised manuscript in Lines 24. The same changes also be done throughout the whole manuscript.

 Comment 5: What is the targets of sediment transport in line 32-34 referred to?

Responds: the Yellow River was the most sediment-laden river in the world, but its sediment transport has decreased by approximately 90% over the past 60 years (Wang et al., 2016). The rapid decrease of sediment discharge to the sea and the increase of grain size of suspended sediments not only changed the sediment dispersal pattern at the estuary, but also modified the shoreline and subaqueous slope (Wang et al., 2010). The sediment transport is one of the important ecological objectives, which is guaranteed by the improvement of e-flows.  

To make this point more clear, we add the sentence “The reduction of the runoff decreases the sediment transport to the sea and changes the sediment dispersal pattern at the estuary [21], and also causes ecological problems.” in the revised manuscript in Lines 100-102.

References:

Wang S, Fu B, Piao S, et al. Reduced sediment transport in the Yellow River due to anthropogenic changes [J]. Nature Geoscience, 2016, 9(1): 38-41.

Wang H, Bi N, Saito Y, et al. Recent changes in sediment delivery by the Huanghe (Yellow River) to the sea: causes and environmental implications in its estuary [J]. Journal of Hydrology, 2010, 391(3-4): 302-313.

 Comment 6: In some places, compensation needed for flow maintenance is presented, but the explanation is very unclear, for example, lines 38-41.

Responds: Thanks for your comments and suggestions. To make this question more clearly, the sentence in Lines 38-41 in the original manuscript To achieve a higher ecological protection objective, economic measures could be used to compensate irrigation stakeholders for agricultural output losses beyond the acceptable economic losses, and 7.1 billion yuan will be paid for e-flows maintenance in dry years.has been revised to “In dry years, more water should be allocated to improve the e-flows. The economic losses may beyond the acceptance of irrigation stakeholders. Then 71×108 yuan will be paid for them to compensate the losses.” in the revised manuscript in Lines 28-30.

 Comment 7: Some typo errors, such as in line 103.

Responds: we have revised the typo errors, and examined the whole manuscript to make sure the similar mistakes would not occur.   

 Comment 8: Different modules are presented, for example, lines 150, but such important parts are embedded in the middle of a paragraph, and readers may encounter a hard time finding them. How about making subtitles for different modules?

Responds: Different modules in Lines 150-166 have been presented by making subtitles for different modules just as the reviewer’s suggested. Detailed revision please refer the revised manuscript.  

 Comment 9: Some writings need further clarity, for example, line 253.

Responds: We polished our manuscript sentences for further clarity. The sentence in Lines 253 in the original manuscript However, the allocation of Yellow River diversion resources cannot be separated from the overall coordination of administrative administrators.has been revised to “However, the implementation of Yellow River water distribution schemes cannot be separated from the overall coordination of administrators.” in the revised manuscript in Lines 192-193.

 Comment 10: In line 255 -257: How about just simplifying “Thiessen interpolation and geostatistical analysis methods within ArcGIS 10.2 are used to calculate agricultural water needs for different districts”.al

Responds: just as the reviewer’s suggested, the sentence in Line 255-257 in the original manuscript “On the ArcGIS 10.2 platform, the Thiessen interpolation method and geostatistical analysis method were used to calculate agricultural water requirements for different regulation districts” has be simplified as “Thiessen interpolation method and geostatistical analysis methods within ArcGIS 10.2 were used to calculate  for different districts.” in the revised manuscript in Line 195-198. Similar modifications has been done though the whole manuscript.

 Comment 11: Some sentences such as lines 304-306 are not clear.

Responds: crop coefficient mentioned in the sentences in Lines 304-306 is a parameter in the equation (4) in Lines 234-235. The sentence “Crop coefficients for different crop growth stages for the Yellow River irrigation area in Shandong Province were obtained from the research of Chen et al. (1995) [27]” has been clarified to “Crop coefficients data for crops planted in the study area were obtained from the research of Chen et al. (1995) [31]” in the revised manuscript Lines 272-273.

 Comment 12: What does “per year for years” refer to such as in line 314-315.

Responds: it is a grammar mistake. We have revised “for years” to “in recent years”.

 Comment 13: Many sentences are run on, for example lines 319-323, very confusing.

Responds: to make it more clear and readable, the sentences in Line 319-326 in the original manuscript The industrial and domestic water requirements from 1998-2017 were obtained from the Yellow River Water Resources Bulletin [18], and the total industrial and domestic water consumption accounted for 7.4%, 14.3% and 15.7% of the annual water diversion resources from the Yellow River by Shandong Province in the 1990s, 2000s and 2010s, respectively. The industrial and domestic water requirements from 1951-1997 and 2018 were determined by the annual average difference in runoff, which was monitored at the Gaocun and Lijin hydrological stations during the same period and multiplied by 7.4% and 15.7%, respectively.has been revised to “The industrial and domestic water requirements data from 1998-2017 were obtained from the Yellow River Water Resources Bulletin [22]. There is no recorded official data for the years 1951-1997 and 2018.  We supplemented the missing data by multiplying the proportion coefficients by the available water resources. The average proportion coefficient was 7.4% for the years 1951-1997, and 15.7% for the year 2018 [22]. The available water resources were the differences of runoff data monitored at Gaocun and Lijin hydrological stations.” in the revised manuscript in Line 285-291.  

 Comment 14: There are many run-on sentences that do not convey the real meaning.

Responds: we revised the whole manuscript, to make sure all the sentences could convey the real meanings. The detailed revision has been showed in the revised manuscript.

Round 2

Reviewer 1 Report

Thank you for the revised version of the manuscript.

Author Response

thanks for reviewer's comments and help.

Reviewer 2 Report

We are not quite understand the meaning of “aerial unit”. Do you refer the unit “billion m3” in the original manuscript in Lines 29-32? I adopted the scientific notation method to represent the large numbers in the revised manuscript. For example “14.8 and 16.8 billion m3” in the original manuscript in Lines30 has been revised to “148 and 168×108 m3” in the revised manuscript in Lines 24. The same changes also be done throughout the whole manuscript.

There is no need to explain the universally used scientific notations. It is well understood.  The question asked was about how much area does this amount of water (148 and 168 x 108 m3) serve?

Rest of the paper is fine after cleaning these corrections.  

Author Response

Comment 1: The question asked was about how much area does this amount of water (148 and 168 x 108 m3) serve?

Responds: the environmental flows are recommended for meeting ecological objectives for the Yellow River Estuary. The freshwater wetland area of the estuary is 792.7 km2 [24]. So the amount of water (148 and 168 x 108 m3) could meet a certain level of ecological objectives for an area about 792.7 km2. We supplemented this information “for the Yellow River Estuary” “The Yellow River Estuary is located in eastern Shandong province. The freshwater wetland area of the estuary is 792.7 km2 [24].” and “in the Yellow River Estuary” in the revised manuscript in Line 26, 96-97 and 104-105.

References:

Sun, T.; Yang, Z.F.; Cui, B.S. Critical environmental flows to support integrated ecological objectives for the Yellow River Estuary, China. Water Resources Management 2008, 22, 973-989.

Other revisions:

  1. We made someimprovement in the results part.

For example, the sentences “In dry years, the total crop output value was 227×108 yuan when the initial e-flows could meet the ecological objective of estuary evaporation consumption and salinity balance. It accounted for 36.74% of the potential crop output value (maximum yield production without water stress) for the Shandong irrigation area.” in the original manuscript in Line 337-340 has been revised to “In dry years, if the initial e-flows could meet the ecological objective of estuary evaporation consumption and salinity balance, we should allocate 134.21×108 m3 water to the Yellow River Estuary. The total crop output value was 227×108 yuan, which accounted for 36.74% of the potential crop output value (maximum yield production without water stress) for the Shandong irrigation area.” in the revised manuscript in Line 337-341.

The detailed revision please refers the revised manuscript.

    1. We have had the manuscript polished by a professional assistance (AJE) to improve our writing. The detailed revision please refers the revised manuscript. The certificate was attached as follows. 
  •  
